# Determinants of Acute Otitis Media in Children: A Case-Control Study in West Java, Indonesia

**DOI:** 10.3390/medicina61020197

**Published:** 2025-01-23

**Authors:** Arif Dermawan, Bejo Ropii, Lina Lasminingrum, Wijana Hasansulama, Budi Setiabudiawan

**Affiliations:** 1Department of Otorhinolaryngology-Head and Neck Surgery, Faculty of Medicine, Universitas Padjadjaran, Kota Bandung 40161, Indonesia; lina.lasminingrum@unpad.ac.id (L.L.); wijana@unpad.ac.id (W.H.); 2Preanger Institute for Biomedical Engineering, Science, and Technology, Depok 16424, Indonesia; 23219348@std.stei.itb.ac.id; 3Department of Child Health, Faculty of Medicine, Universitas Padjadjaran, Dr. Hasan Sadikin General Hospital, Kota Bandung 40161, Indonesia; budi.setiabudiawan@unpad.ac.id; 4Faculty of Medicine, President University, Cikarang, Bekasi 17550, Indonesia

**Keywords:** acute otitis media, allergic rhinitis, smoking, nutrition, immunization, pediatrics

## Abstract

*Background and Objectives*: Acute Otitis Media (AOM) is a leading cause of morbidity in children, characterized by fever, otalgia, and hearing loss. If untreated, AOM may progress to chronic complications requiring surgical management. Globally, factors such as allergies, environmental tobacco smoke, and nutritional deficiencies are well-established risk factors, but in Indonesia, particularly rural areas like Bandung Regency, limited awareness and data exacerbate the burden of disease. Smoking prevalence and low immunization rates further increase risks for AOM. *Materials and Methods*: This case-control study, conducted between September 2019 and February 2020 in Bandung Regency, evaluated risk factors for AOM in children aged 24–59 months. Data were collected through structured questionnaires, anthropometric assessments, and ENT examinations. A total of 168 AOM-positive and 367 AOM-negative children were recruited from primary healthcare facilities. *Results*: Multivariable analysis identified significant associations with AOM: allergic rhinitis (AOR 1.92), cigarette smoke exposure (AOR 1.79), stunted growth (AOR 1.48), and incomplete basic immunizations (AOR 1.77). These findings highlight the importance of addressing modifiable factors such as nutrition and immunization to reduce AOM incidence. *Conclusions*: The rhinitis allergy and exposure to cigarette smoke are among the well-established risk factors that our results validate. Additional research is necessary to validate if our findings involving two modifiable risk factors, stunted children and insufficient basic vaccination, may increase the risk of AOM.

## 1. Introduction

Acute otitis media (AOM) is a prevalent infection in early childhood, often arising as a complication of upper respiratory tract infections. It is characterized by middle-ear effusion and the acute onset of symptoms caused by middle-ear inflammation, including earache in older children or non-specific signs such as fever, irritability, or poor feeding in younger children. Key indicators include a distinctly dark, yellow, or cloudy tympanic membrane, often accompanied by bulging, air–fluid levels, perforation, or ear discharge. Diagnostic tools like pneumatic otoscopy and tympanometry help confirm the presence of middle-ear effusion. In children with grommets, ear discharge indicates AOM, as fluid from the middle ear drains into the ear canal. Common bacterial causes include *Streptococcus pneumoniae*, non-typeable *Haemophilus influenzae*, and *Moraxella catarrhalis*, with evidence suggesting a shift in predominance from *Streptococcus pneumoniae* to non-typeable *Haemophilus influenzae* following pneumococcal vaccination [1,2,3].

Acute Otitis Media (AOM) is a common middle ear condition characterized by symptoms such as fever, pain, hearing loss, and discharge [4]. It is the second most frequently diagnosed pediatric condition in the emergency room, following upper respiratory infections [5]. Acute otitis media (AOM) is one of the most commonly diagnosed conditions worldwide. According to the Centers for Disease Control and Prevention (CDC), its prevalence in the United States has risen dramatically, with a reported increase of 150%. By the age of two, around 70% of children will have had at least one episode of AOM. Studies show that AOM is a major contributor to the use of empiric antibiotics in the U.S. Otitis media is classified into different types, depending on the severity of symptoms and the presence of complications [6,7].

Untreated AOM may progress to chronic suppurative otitis media which requires surgery [8]. AOM is caused by Eustachian tube dysfunction, which obstructs fluid flow from the middle ear cavity. The static middle ear’s fluid serves as a favorable environment for pathogen development. Several studies reported the risk factors associated with AOM such as allergy, parental smoking, breastfeeding, pacifier use, upper respiratory infection, and family history. Risk factors such as group childcare and passive smoking also significantly contribute to AOM occurrence [1,5,7].

Risk factors for chronic otitis media (COM) and recurrent otitis media (ROM) include snoring, previous episodes of AOM/ROM, second-hand smoke exposure, and low socioeconomic status. Further research links increased AOM recurrence to factors such as low parental education, exposure to smoke or mold, laryngopharyngeal reflux disease, lack of breastfeeding, allergies, and ongoing respiratory symptoms like cough and rhinorrhea [9].

Recurrent otitis media (ROM) is diagnosed when a child has three or more separate episodes of acute otitis media (AOM) within six months, or at least four episodes within a year. A more persistent and complex type is otitis media with effusion (OME), which is classified as chronic otitis media [10,11]. OME is characterized by fluid trapped in the middle ear cavity without the acute symptoms typically associated with AOM. Surgical procedures like tympanostomy tube insertion and myringotomy are the most commonly performed interventions to manage these conditions and alleviate related complications [10,11,12].

The field of otolaryngology focuses on the anatomical, physiological, and pathophysiological aspects of the ear, nose, and throat, providing essential insights for the effective management of AOM. A comprehensive understanding of these areas is crucial for advancing treatment strategies [6].

AOM was notably common in Indonesia, particularly in rural areas where the majority of cases are concentrated [13]. Acute otitis media (AOM) remains an under-recognized health concern in Indonesia, partly due to low community awareness. A study conducted in Surabaya highlighted the issue by analyzing chronic suppurative otitis media (CSOM) cases in 38 children using rigorous microbiological methods. Despite its prevalence, data on AOM risk factors in children are sparse [14].

In 2016, Bandung, West Java, faced alarming public health challenges due to high smoking rates. A local survey revealed that 37% of residents aged 16 to 49 were tobacco smokers, with over 41% of these individuals having smoked for more than a decade. Notably, 90% of these smokers were part of a “smoking habit group”, emphasizing smoking as a socially normalized behavior. This data highlights the deeply entrenched cultural acceptance of smoking within Bandung, posing significant barriers to tobacco control and public health interventions [15].

However, the information on the risk factors for AOM in children, particularly in Indonesian rural area is limited. This study aimed to look into the risk factors for AOM among children in Bandung Regency, Indonesia. The findings of this investigation will provide key factors of AOM occurrence as valuable information for preventing AOM.

## 2. Materials and Methods

### 2.1. Study Design and Participant Selection

Case-control research was performed to identify the risk factors linked with the onset of AOM in children aged 24 to 59 months in Bandung Regency, Indonesia. The research subjects were selected using a multistage random sampling method, adhering to predetermined research criteria. Parents or guardians who agreed to participate provided informed consent after receiving a thorough explanation of the study. The sampling process was conducted in several stages, with each stage progressively narrowing the sampling units. In the first stage, 11 subdistricts were selected from the 31 subdistricts in Bandung Regency. Next, 39 villages were chosen from the 81 villages within the selected subdistricts. In the third stage, 29 community health centers (puskesmas) were selected from 58 puskesmas in the identified villages. This was followed by a fourth stage, where 30 puskesmas were chosen from the 58 available. Finally, 800 children were selected from a total of 1544 children registered at the 30 puskesmas.

### 2.2. Data Collection

A multi-stage random sample technique was used to choose primary health care providers in Bandung Regency, Indonesia, between September 2019 and February 2020. After obtaining parental or guardian approval, structured interviews using a questionnaire were done to collect information about sociodemographic information. These data were obtained from the ALG study conducted in Bandung Regency, Indonesia.

In this study, nutritional status was re-evaluated by conducting anthropometric measurements on the subjects, resulting in two groups: children with stunted growth and children with normal growth. Parents or guardians of subjects who consented to participate in the study were interviewed using the OMA questionnaire, while the child subjects underwent a comprehensive ENT examination and tympanometry to determine the presence or absence of acute otitis media (OMA). Data collection for case and control groups meeting the research criteria was carried out through interviews and questionnaires. Additionally, age and sex were used as matching criteria.

### 2.3. Otorhinolaryngology Physical Exam

All study participants underwent otorhinolaryngology physical examinations, which included a comprehensive inspection of the external ear, tympanic membrane, and nasal passages to detect any abnormalities or evidence of an ear infection. The clinical examination revealed abnormalities in the tympanic membrane, characterized by the absence of the light reflex and the presence of fluid. Supporting diagnostic tests showed tympanometry results with a Type B or As curve, further indicating dysfunction.

### 2.4. Ethical Considerations

This study was approved by the Ethics Committee of the Padjajaran University with approval number 1170/UN6.KEP/EC/2019 and approval date 12 September 2019. The children’s parents or legal guardians provided informed consent, and the acquired data was kept fully confidential throughout this study.

### 2.5. Statistical Analysis

The data were analyzed using bivariate analysis with the Chi-squared test to assess the relationship between independent and dependent variables, and multiple logistic regression analysis to examine factors associated with the incidence of AOM using SPSS software version 25 (IBM SPSS, Armonk, NY, USA). Factors with a *p*-value <0.25 from the bivariate analysis were included in the multivariable analysis. Stepwise removal was performed to refine the model, retaining only statistically significant variables. To address missing data, this approach aimed to optimize the sample size. Sensitivity analysis was conducted by reintroducing all eliminated variables into the final model to confirm they remained statistically insignificant.

## 3. Results

All children who attended the selected primary healthcare facilities during the study period were included in the study. A total of 168 AOM-positive and 367-AOM negative children were enrolled. Figure 1 shows how participants were selected. All variables were fully curated from all participants.

Our study showed no significant age differences observed between case and control groups. A bivariate analysis test was conducted to test the correlation between variables with the incidence of AOM (Table 1). Variables with *p*-values over 0.25 univariably do not correlate with the occurrence of AOM in children aged 24–59 months in Bandung Regency. Meanwhile, six variables with *p*-values less than 0.25, including allergic rhinitis, non-allergic rhinitis, nutritional status, exposure to cigarette smoke, nutritional status, complete basic immunization, and breastfeeding position, were included in further multivariable analysis (Table 2).

In the final multivariable model (Table 2), four of the six factors have significant correlation (*p*-value less than 0.25) with the incidence of AOM in children between the ages of 24 and 59 months. Compared to those without allergic rhinitis, allergic rhinitis children had a 1.92-fold increase chance of developing AOM. The occurrence of AOM is linked to cigarette smoke exposure, with a risk 1.79 times higher than in the non-smoke-exposed group. Stunted children have a 1.48 higher chance of having AOM than normal children based on weight measurements and nutritional status variables. The risk of acquiring AOM is 1.77 times greater in children who do not receive the full course of basic immunizations than in those who have.

## 4. Discussion

Our study showed that no significant age differences were observed between the case and control groups. A bivariate analysis test was conducted to test the correlation between variables with the incidence of AOM (Table 1). Variables with *p*-values over 0.25 univariably do not correlate with the occurrence of AOM in children aged 24–59 months in Bandung Regency. Acute otitis media (AOM) is often overlooked and underreported in Indonesia, with limited epidemiological data and public awareness of its prevention. A previous study by Wijayanti highlights a 4.64% AOM prevalence in Banyumas Regency, based on clinical examinations in primary schools by ENT specialists. This rate surpasses previous findings in North Sumatra (2.1%) and a study by Anggraeni et al. (2019) involving 7005 schoolchildren (2.5%), emphasizing the need for improved reporting, awareness, and prevention efforts [1,16].

Apart from established risk factors, including allergic rhinitis, exposure to cigarette smoke, and lying down when breastfeeding, we also found certain modifiable risk variables connected to AOM. These included stunted growth and incomplete basic vaccination [1,5,7].

In this study, allergic rhinitis was the most influential variable related to AOM in children ages 24–59 months. It is supported by the previous case study in Parakou and Sanglah Denpasar on kids ages 0–12 years. Hounkpatin et al. highlighted a significant association between recurrent rhinitis and acute otitis media (AOM). Their study found that children with chronic rhinitis had a 5.13 times higher risk (95% CI: 1.31–3.04) of developing AOM compared to those without rhinitis. The persistence of rhinitis likely facilitates the migration of bacteria from nasal secretions and the nasopharynx into the tympanic cavity via the Eustachian tube. Supporting this, a 2010 epidemiological study in South India among preschool-aged children identified persistent rhinorrhea, affecting 10% of children in the region, as a critical risk factor for AOM development. This underscores the importance of managing chronic rhinitis to reduce AOM incidence [17,18].

Another meta-analysis study found that allergic rhinitis was a notable risk factor for otitis media with effusion [19]. Allergic reactions can trigger the production of cytokines and other inflammatory mediators in nasal passage. This inflammation can spread to the tissues around the Eustachian tube, then block the air and fluid movement in and out the middle ear. The airflow interruption increases nitrogen absorption and causes chronic negative pressure, leading to the development of otitis media. Furthermore, the persistence of rhinitis increases the probability of bacteria migration in the nasal cavity to the tympanic cavity, causing concurrent significant AOM [17,20].

Three well-established indicators of children’s nutritional status are underweight, stunting, and wasting [21]. The symptoms of stunting include lower body weight, delayed bone growth, and shorter height or body length than the average children in their age group. According to Handryastuti et al. (2022), stunted children are more susceptible to disease due to poor nutrition, frequent infections, and insufficient psychosocial stimulation. The malnourished body has a weakened immune system, increasing the chance of developing AOM [17]. A case study in Parakou revealed that children that have severe acute malnutrition (SAM) were about 2.23 times more likely and children with moderate acute malnutrition (MAM) were twice as likely to develop AOM than children in good nutritional status. Similarly, a study in Lucknow on 851 participants aged 1–5 years revealed a significant positive correlation (*p* < 0.001) between malnutrition and otitis media [22].

This study found that nutritional status is correlated with the incidence of AOM, with poor nutritional status often reflecting low economic conditions in developing countries, which increases susceptibility to inflammatory ear diseases. Evidence from prior research highlights that deficiencies in zinc, iron, vitamin A, and vitamin D are associated with AOM. However, this study assessed nutritional status solely through anthropometric measurements (weight-for-age), and further research into specific macro- and micronutrient deficiencies is needed [1,17].

In 2019, half of the children in Bandung Regency were exposed to tobacco smoke. Thus, understanding the association between children’s exposure to cigarette smoke and the occurrence of AOM is critical. Our findings on children exposed to cigarette smoke are consistent with the findings of Athbi and Abed-Ali (2020), which shows that 84% of newborns exposed to secondhand smoke had a significant association with having AOM [23].

Parental smoking has been linked to an increased risk of AOM and worsening of its symptoms, according to a different study done in Iran on 250 children aged 1–12. Passive smoking significantly contributes to the development of otitis media (OM) in children by facilitating bacterial adherence to the respiratory epithelium, suppressing local immune responses, and impairing mucociliary clearance. These effects create a favorable environment for bacterial proliferation and infection in the middle ear, increasing the risk of OM [24]. Other studies also demonstrated that parental or passive smoking have been linked to the development of AOM [23,25].

Passive smoking after birth causes ciliostasis, goblet cell hyperplasia, mucus hypersecretion, and vascular congestion, which may result in the accumulation of mucus and bacteria in the middle ear, resulting in AOM [26]. Chemical irritation indirectly causes bacterial and viral illness by blocking the ear canal or altering the immune system. In contrast with our findings, previous research did not find any correlation between AOM and parental smoking [27].

Instead of parental smoking, children with a history of upper respiratory tract infections were more likely to develop otitis media, though its underlying cause remains debated, potentially involving both genetic and environmental factors. Previous studies have identified genetic associations with OM, including interleukin (IL) genes, mucin genes, TLR4, FBXO11, and TNFα. Beyond genetics, shared living environments may contribute to the correlation, as family members are often exposed to similar risk factors [28].

We found that parental education does not correlate with the incidence of AOM in children in this study. Regarding parental education, the results of this study differ from the findings by Martines et al., which stated that upper respiratory tract infections and otitis media are common childhood diseases strongly associated with low parental educational attainment. However, this research aligns with findings from previous studies regarding the correlation of several risk factors contributing to AOM in children, such as the lack of breastfeeding (*p* = 0.0014). Additionally, an increased risk of recurrent otitis media has been observed in cases involving allergies, persistent cough, and runny nose (*p* = 0.0001) [9].

Several research findings suggest that exclusive breastfeeding can reduce the risk of AOM [29]. Breastfeeding exclusively throughout the first 6 months of birth was linked to a 43% decrease in AOM in the first 2 years of life [30]. Human breast milk contains secretory immunoglobulin A, which offers targeted immunological defense against infections on all mucosal surfaces by impeding the microorganism’s ability to adhere to surfaces [31]. Our study, which focused on the ages of 24–59 months, revealed no association between AOM incidence and the absence of exclusive breastfeeding. This is probably because breastfeeding does not protect against AOM after the age of 2 years [32].

Our investigation showed that the breastfeeding position did not correlate with AOM incidence, which aligns with a previous study by Nadal et al. (2017). Because nursing requires a significant energy expenditure and is dependent on the mother’s physical condition, it is inevitable that nursing may cause moments of exhaustion [33]. Thus, in practice, the majority-selected position is the most comfortable for both the mother and their children. However, it is strongly advised that the infant’s head and shoulders be raised and supported, if possible, by an arm or a pillow. A previous study showed that AOM can also occur more frequently in infants who are breastfed incorrectly. A study in Pakistan showed that only the supine posture and the incidence of AOM were significantly correlated [34]. Other research also demonstrated that AOM in children is significantly elevated when they are fed in the supine posture because this position may causes milk to flow into the ear canal, increasing the likelihood of respiratory bacteria to enter the middle ear, which can result in AOM [23,35,36].

When a child is younger than one year old, they should receive the full basic immunization to prevent numerous illnesses [37,38]. Studies on the connection between AOM and the childhood vaccinations are scarce. Nonetheless, other research indicates that some vaccines, including the conjugate pneumococcal and regular influenza vaccinations in children and newborns, have contributed to a decline of AOM frequency [39,40,41]. Accordingly, our study provides new information about the correlation between children not receiving all recommended vaccinations and the risk of AOM. This information can be used by health professionals as a reference to help parents understand the importance of their children receiving their basic immunizations.

Our study found no link between pacifier use and the prevalence of AOM in children aged 24–59 months. This contradicts the previous findings, which suggest that there is an association between extended use of pacifiers and the occurrence of AOM in children under four years old [42,43,44]. Another case study shows the use of pacifiers linked to ear infections due to the possibility of throat bacteria migrating from the auditory canal to the middle ear, causing otitis media [23,45,46]. Parents must keep pacifiers clean for their kids to use when necessary.

While developed nations identify different AOM risk factors—such as childcare, allergies, smoking, and socioeconomic status—this study emphasizes the significance of region-specific lifestyle and cultural habits, particularly in rural Indonesia. Identifying such localized risk factors is crucial for designing effective prevention programs tailored to the community.

## 5. Conclusions

The rhinitis allergy and exposure to cigarette smoke are among the well-established risk factors that our results validate. Additional research is necessary to validate if our findings involving two modifiable risk factors, stunted children and insufficient basic vaccination, may increase the risk of AOM. Acquiring knowledge about these variables is crucial for implementing measures to mitigate and stop the occurrence of AOM in children, specifically aged 24–59 months. Policies must be developed in cooperation with various stakeholders, particularly policy-making authorities, and thereafter disseminated to healthcare providers and the Bandung district community.

This study was conducted in a rural region, which may limit the generalizability of the findings. The unequally distributed proportion of socioeconomic backgrounds in our study could also bias the results. This study was conducted over a six-month period, which may not capture seasonal variations in the incidence of AOM or related exposures (e.g., respiratory infections). If certain risk factors fluctuate throughout the year, the associations observed during this limited timeframe may not fully represent the year-round risk profile for AOM. Therefore, longer sampling is encouraged.

## Figures and Tables

**Figure 1 medicina-61-00197-f001:**
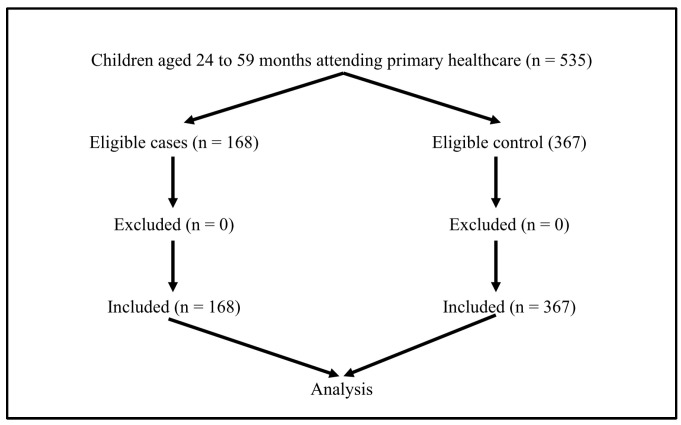
Flow diagram of participant selection process.

**Table 1 medicina-61-00197-t001:** Relationship between risk factors and the incidence of AOM in children aged 24–59 months in Bandung Regency, Indonesia.

Risk Factor	AOM		*p* Value	OR (95% CI)
	Positive (*n* = 168)	Negative (*n* = 367)		
Age (month)			0.314	
24–35	56 (33.3)	100 (27.2)		1.26 (0.81–1.98)
36–47	53 (31.5)	134 (36.5)		0.89 (0.57–1.39)
48–59	59 (35.1)	133 (36.2)		1.0
Gender:			0.269	
Male	82 (48.8)	198 (54.0)		0.81 (0.56–1.17)
Female	86 (51.2)	169 (46.0)		
Allergic rhinitis:			0.002 *	
Yes	66 (39.3)	93 (25.8)		1.86 (1.26–2.75)
No	102 (60.7)	268 (74.2)		
Non-allergic rhinitis:			0.046 *	
Yes	92 (54.8)	164 (45.4)		1.45 (1.01–2.10)
No	76 (45.2)	197 (54.6)		
Cigarette smoke exposure:			0.021 *	
Present	145 (86.8)	283 (78.4)		1.82 (1.09–3.04)
Absent	22 (13.2)	78 (21.6)		
Family’s income/month (million Rp):			0.959	
<1.5	111 (67.7)	238 (66.1)		1.17 (0.22–6.10)
1.5–2.5	40 (24.4)	95 (26.4)		1.05 (0.20–5.65)
1.5–2.5	11 (6.7)	22 (6.1)		1.25 (0.21–7.50)
>3.5	2 (1.2)	5 (1.4)		1.0
Nutritional status:			0.023 *	
Stunted	102 (60.7)	184 (50.1)		1.54 (1.06–2.23)
Normal	66 (39.3)	183 (49.9)		
Exclusive breastfeeding:			0.537	
Yes	160 (95.2)	339 (93.9)		1.30 (0.57–2.98)
No	8 (4.8)	22 (6.1)		
Complete basic immunization:			0.018 *	
No	33 (19.6)	43 (11.9)		1.81 (1.10–2.97)
Yes	135 (80.4)	318 (88.1)		
Father’s education:			0.666	
Elementary School	67 (39.9)	127 (34.7)		1.58 (0.42–6.04)
Junior High School	53 (31.5)	129 (35.2)		1.23 (0.32–4.73)
Senior High School	45 (26.8)	101 (27.6)		1.34 (0.34–5.17)
College	3 (1.8)	9 (2.5)		1.0
Mother’s education:			0.577	
Elementary School	64 (38.1)	125 (34.1)		0.90 (0.25–3.17)
Junior High School	60 (35.7)	133 (36.3)		0.79 (0.22–2.80)
Senior High School	40 (23.8)	101 (27.6)		0.69 (0.19–2.50)
College	4 (2.4)	7 (1.9)		1.0
Breastfeeding position:			0.124	
Lying	44 (26.2)	73 (20.2)		1.40 (0.91–2.15)
Sitting	124 (73.8)	288 (79.8)		
Use of pacifiers:			0.262	
Yes	127 (75.6)	256 (70.9)		1.27 (0.84–1.93)
No	41 (24.4)	105 (29.1)		
Nasal voice:			1.00	
Yes	1 (0.6)	4 (1.1)		0.53 (0.06–4.82)
No	167 (99.4)	357 (98.9)		

* Based on Chi-squared test. OR (95% CI): odds ratio (95% confidence interval).

**Table 2 medicina-61-00197-t002:** Multivariable analysis of factors associated with the incidence of AOM in children aged 24–59 months in Bandung Regency, Indonesia.

Variable	Coeff B	SE (B)	*p* Value	OR Adj (95% CI) *
Allergic rhinitis	0.654	0.281	0.020	1.92 (1.10–3.34)
Non-allergic rhinitis	−0.081	0.270	0.765	0.92 (0.54–1.57)
Cigarette smoke exposure (present)	0.582	0.267	0.029	1.79 (1.06–3.02)
Nutritional status (stunted)	0.389	0.196	0.047	1.48 (1.01–2.17)
Incomplete basic immunization	0.572	0.260	0.028	1.77 (1.06–2.95)
Breastfeeding position (lying)	0.217	0.231	0.348	1.24 (0.79–1.96)

* OR adj (odds ratio adjusted).

## Data Availability

The data supporting the findings of this study are available from the corresponding author upon reasonable request.

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
