# Peer review of "Determinants of Acute Otitis Media in Children: A Case-Control Study in West Java, Indonesia"

_medicina, 2025, doi:10.3390/medicina61020197_

Round 1

Reviewer 1 Report

Comments and Suggestions for Authors

I thank the Editors for selecting me as a reviewer for this article. The article aimed to investigate the risk factors for Acute Otitis Media (AOM) among children in Bandung Regency, Indonesia. The findings of this study provide key factors related to AOM occurrence, offering valuable information for its prevention.

The article addresses the risk factors of a very common condition in pediatric age, namely AOM. The sample includes 168 AOM-positive and 367 AOM-negative children, collected between September 2019 and February 2020.

I believe that the article does not provide any new information compared to what is already known and covers a sample from only six months.

Although it is interesting that the study was conducted in a rural region, I think a larger sample size is necessary for publication. For example, data on children affected by AOM could be collected in the subsequent months up to 2024 (dividing the colder months from the warmer ones).

Author Response

Thank you for your insightful comment and input. Here are our responses:

  • Comments 1: Enrich the contents: To facilitate transparent and open science, we encourage authors to publish their results and experimental methodology in as much detail as possible so that results can be reproduced. We noticed that the main text of your manuscript is quite brief which may mean that the materials and methods, research background, future research directions, or possible applications of the research are not described in enough detail. Please consider the following points in your revisions: adding full experimental details, presenting completely all the results, and describing a comprehensive background to the research in the introduction section.

Response 1: We have carefully revised the document to address the concerns raised. In response to your comments, we have enriched the content by adding more details to the experimental methodology, ensuring that all steps, tools, and parameters are clearly described to facilitate reproducibility. The results section has also been expanded to present all findings comprehensively, accompanied by detailed explanations. Furthermore, we have enhanced the introduction by including a more thorough research background, providing better context and justification for our study. In the discussion section, we have interpreted statistically significant results in the context of existing research, highlighting the implications of our findings and their contribution to the field. We have also addressed potential limitations and outlined future research directions, as well as possible applications of our study to broaden its relevance and impact. The experimental methodology section has been clarified and detailed further to ensure transparency and reproducibility.

  • Comments 2: You are required to provide the date and ethics code.

Ethic Committee Name: the Institutional Ethics Committee of Padjajaran

University, Indonesia

Approval Code:

Approval Date:.   

Response 2: We have added the information regarding the approval date and ethics code from the Institutional Ethics Committee of Padjajaran University, Indonesia in the methodology section (page 3). This includes the approval code and date, as requested.

  • Comments 3: We noticed that some sections had significant overlap with previously published articles, and we request you to rephrase the highlighted parts.

Response 3: We appreciate this comment and agree with it. We have carefully revised the manuscript and rephrased the highlighted sections to address the issue of overlap with previously published articles. These adjustments were made to minimize plagiarism while maintaining the clarity and integrity of the content.

  • Additional revision à Plagiarism 32%.

Response: Thank you for your feedback. We have rephrased the highlighted sections in accordance with the revision points provided on Turnitin. These adjustments were made to minimize plagiarism while ensuring the clarity and accuracy of the content.

We hope this revision addresses your concerns.

Reviewer 2 Report

Comments and Suggestions for Authors

Dear editor

Thanks a lot for hard work. I read this article with interest. However, I have some concerns.

Kindly incorporate the responses within the manuscript to augment its overall quality.

In the introduction and discussion, the authors should diversify acute otitis from Otitis Media with Effusion. and give other reasons for acute inflammation in children such as tonsil hypertrophy, anatomical conditions and developmental defects, etc.

In the method they should determine.

In the method, the authors should determine the criteria for switching on and excluding from the test groups.

In the discussion, they should define work restrictions.

Conclusions should be more detailed and measures for medical facts arising from work.

Literature should be supplemented with DOI: 10.3390/ijerph18073555.

Comments on the Quality of English Language

There are many spelling and language mistakes and the manuscript needs to be corrected by a native English speaker.

Author Response

(The authors gave the same response as above.)

Reviewer 3 Report

Comments and Suggestions for Authors

This study investigated the association between acute otitis media (AOM) and several factors, including age, gender, allergic rhinitis, tobacco smoke exposure, family income, nutritional status, breastfeeding, basic immunizations, family education, and nasal voice, and found that allergic rhinitis, tobacco smoke exposure, nutritional status, and non-completion of basic immunizations were the most common factors related to AOM. response factors regarding AOM. The data in this study are not new and will add small insights into previous studies.

Was there any statistical evaluation and discussion of variance inflation factor?

Because of the large number of variables, an evaluation of multicollinearity may be necessary.

Minor concerns:

Double periods (L179).

The appendix contents should be completed (L275-L284).

Author Response

(The authors gave the same response as above.)

Round 2

Reviewer 1 Report

Comments and Suggestions for Authors

I thank the authors for the revisions made. Although the article is not particularly innovative and includes a small number of patients, considering the high prevalence of the condition, I believe that, in this form, it is clearer and more engaging, even within a six-month timeframe. However, I believe the article should be reviewed by a native English speaker to improve the fluency of the text.

Comments on the Quality of English Language

the article should be reviewed by a native English speaker to improve the fluency of the text

Reviewer 2 Report

Comments and Suggestions for Authors

Accept in present form